# Social Media Marketing for Small and Medium Enterprise Performance in Uganda: A Structural Equation Model

**Cliff R. Kikawa** [1,*], **Charity Kiconco** [2], **Moses Agaba** [2], **Dimas Ntirampeba** [3], **Amos Ssematimba** [4] **and Billy M. Kalema** [5]

1 Department of Economics and Statistics, Kabale University, Kabale P.O. Box 317, Uganda
2 Department of Business Studies, Kabale University, Kabale P.O. Box 317, Uganda
3 Department of Mathematics and Statistics, Namibia University of Science and Technology, 13 Jackson Kaujeua Street, Windhoek 10005, Namibia
4 Department of Mathematics, Gulu University, Gulu P.O. Box 166, Uganda
5 School of Computing and Mathematical Sciences, University of Mpumalanga, Mpumalanga, Cnr R40 and D725 Roads, Mbombela 1200, South Africa
* Correspondence: richard.kikawa@gmail.com or crkikawa@kab.ac.ug

**Abstract:** Thanks to the ongoing expansion of internet access and, most recently, the movement restrictions that were put in place globally to stop COVID-19 spread, many small and medium enterprises (SMEs) are prepared to use social media platforms to market their products as a way to improve their business performance in emerging economies. Businesses at all levels that use social media marketing (SMM) see a considerable increase in their output. This study's objective is to identify the factors that significantly help Ugandan SMEs implement SMM techniques to enhance their commercial performance. Here, statistical models are utilized to analyze how the age and gender of SMEs owners as moderating variables affect the adoption and performance of SMEs in Uganda. A theoretical model that is based on the Technology Acceptance Model (TAM) and Innovation Diffusion Theory (IDT) theories has been put out as part of a specific conceptual framework. The following indicators are used to evaluate the performance of SMEs: sales, customer engagement, awareness of customers' needs, low operation costs, and brand modification by staff. Empirical model validation has been performed using 152 business units (observation units) corresponding to the number of respondents (units of analysis), and the ensuing analyses have been done using structural equation modelling (SEM). The results indicate that compatibility and perceived ease of use have a positive impact on SMEs to adopt SMM, while perceived usefulness has a negative impact on SMEs to adopt SMM. Age and gender as moderating variables all have a positive moderating effect. With limited studies available on the subject, this research contributes to already existing literature by combining two components of the TAM model and one component of the IDT to explain the impact of SMM on SMEs when moderated by both age and gender in a developing economy.

**Keywords:** social media marketing; electronic marketing; social media influence; small and medium enterprises (SMEs); structural equation modelling; moderating variables

## 1. Introduction

Small and medium enterprises (SMEs) play an important role in the development of economies and communities [1] (Bajdor, Pawełoszek & Fidlerova, 2021). These collectively comprise about 90% of private sector production, employing over 2.5 million people [2]. The Ministry of Finance, Planning and Economic Development (MFPED) documents that a substantial number of SMEs have less than 20 employees. MFPED defines a 'Small Enterprise' as an enterprise employing between 5 and 10 staff, with an annual sales turnover or total asset of up to Uganda Shillings (USH) 360 million (1 million USD), and a 'Medium Enterprise' is a business unit that employs more than 50 people with an annual sales turnover or assets of between (USH) 360 million and 30 billion (1–83 million USD) [2].

Agriculture makes up a sizable portion of SMEs (14%), followed by education and health (13%), and leisure and personal (10%). SMEs in Uganda are relatively new business entities; a major portion (69%) of them are between one and ten years old [3]. SMEs are run and typically managed by owners. Under a third (31%) have a manager who is in charge of operations [4].

Information and communication technology (ICT) offers a wide range of uses and improves marketing efforts in SMEs [5]. Online social networks (OSNs) have developed over time as one of the applications that have been implemented as a result of the growth of Web 2.0 [6,7]. Social media is considered to be one of the main ICT components with the most significant impact on business among all the apps that are now in use. In recent years, there has been a lot of anticipation and enthusiasm surrounding social media's crucial role as one of the most important virtual channels for engaging with customers [8]. On the basis of this, the researchers felt it was important to investigate if the use of social media can accelerate the growth of SMEs in Uganda. Refs. [9] and [10] document that SMEs in Uganda can perform better overall when they use social media, according to the scant research that has been done on the topic. According to the examined literature, there appears to be a huge knowledge gap on how social media usage might improve the performance of SMEs in Uganda. Addressing this gap in the literature is necessary since insights from Uganda could also be applied to other comparable emerging economies.

Nevertheless, different scholars, researchers, and academics have documented several advantages of using social media in the development of SMEs in the global perspective. By adopting social media platforms, consumers and service providers can directly link up with the latter to showcase their product developments and brands [11]. Social media marketing strategies have given rise to new business models such as "social commerce". It is regarded as a channel to seamlessly enable stakeholders get actively involved in online events and dealings through social media for business prospects, product comparison, selling, and buying transactions in order to make decisions [12]. The gap between SMEs and potential customers is said to have been filled by social media marketing [13,14]. The approach by the firms to operate with the augmentation of social media can be regarded as social media marketing (SMM) [15,16]. It is, therefore, important that a holistic exploration is made to ascertain whether the adoption of SMM by Ugandan business firms significantly influences their performance through increased sales, customer interaction, product innovation [17], and branding. Therefore, this study attempts to propose factors that could significantly influence SMM and to explore whether SMM could act as a purposeful tool for the realistic development of the SMEs in Uganda. In this study, the effects of moderating factors (i.e., age and education) on SMM are also not ignored. Hence, there is an attempt to investigate the research questions below by employing a combination of theories as discussed later in the subsequent sections.

- What are the factors that influence social media marketing in SMEs in Uganda?
- How does social media marketing impact these SMEs' performance when moderated by age and gender?

The rest of this paper is organized as follows: The theoretical development of the proposed model is firstly presented, followed by the research methodology, and study results from the analysis. A discussion of the study contribution to theory and empirical implications of the research findings is then presented. Lastly are the limitations of the current study and future research directions, followed by conclusions.

## 2. Theoretical Background and Hypotheses Development

### 2.1. Theoretical Background

One wonders if Ugandan SMEs would adopt SMM innovations in the age of new technology breakthroughs, which many developing nations have embraced, and what the key factors are that could encourage or otherwise prevent SMEs from embracing SMM technologies. In this regard, there is need for inquiry into the worry of technological adoption. When interest is focused on the acceptance of a technological devel-

opment in a community, usually the Technology Acceptance Model (TAM) that was proposed by [18] Davis (1989) is given priority as it is considered to possess a great influence and is universally an appealing model toward ones' acceptance of a novel technology [13,19,20]. The principal of TAM theory is grounded on two explanatory variables, which are "perceived usefulness (PEU)" and "perceived ease of use (PEOU)". Again, Charness, and Boot (2016) also stated that one of the most popular models of technology acceptance is the Technology Acceptance Model [18], which focuses on the two main elements that affect a person's intention to utilize new technology: the perceived usability and simplicity of use. Given this context, only two of the TAM model's components were employed in this study, acknowledging that the adoption of new technologies depends largely on people's views toward two particular measures (perceived usefulness and perceived ease of use). While perceived ease of use is interpreted as the point at which a person believes that utilizing a certain system would be effortless, the perceived utility is defined as the point at which the intended user believes that using a particular system will increase his or her job performance [18]. The two explanatory variables as theorized by TAM moderate the relationship between exogenous variables, involving system characteristics, development process, training, and the intention to use a system [21–23]. The two components are ones' beliefs in information technology and form his or her attitude towards technology that subsequently predicts acceptance intention to use a new technology [24].

Ref. [25] stated that the relationships are justified by the fact that the technology that is easy to use and essentially useful to an individual will have a positive impact on the individual's attitude and intention toward using the technology.

TAM components, that is PEU and PEOU, entail several other beliefs such as intention and attitude. A number of proposed adoption models have used TAM as a benchmark in the past and are currently doing so. Based on TAM, many other adoption models have been framed subsequently [26]. With changes in the technological advancements over time, it is necessary to propose other demographic, technological, and social factors towards the adoption of SMM in SMEs. In this regard, this work has employed some of the technology adoption models, for example, TAM (PEU and PEOU) and the Innovation Diffusion Theory (IDT) that was proposed by [27] supported by [28]. In the IDT model, a factor of compatibility (COM) is borrowed. "Compatibility" is the level at which an innovation is perceived as dependable with the existing values, past experiences, and needs of those that intend to use it [27]. The study chose to test only a single component (compatibility) of IDT based on work that was done by [29] which points out that, despite the fact that IDT includes more extensive elements, accumulating empirical data have revealed that TAM offers a superior mechanism for describing user acceptance, recognition, and behavior. In other words, factors from TAM (PEU and PEOU) and IDT (COM) have been considered in this study as important predictors of SMM [30,31]. When employees and managers of the SMEs are to accept the communication concerning SMM from relevant channels over an unspecified period of time, they soon realize that indeed this technology aligns well with their previous experience and daily practices with the devices that they already possess, thus they feel compatible to adopt and perhaps implement it in their businesses [13,24]. Uganda, being a developing country with an economy that is struggling to keep afloat, users ever keep themselves aware and cautious of the cost of any innovation that comes onto the market [32,33]. If the adoption and subsequent use of SMM is not affordable relative to the already available practices, there will be a significant hesitation by the SMEs in the adoption of SMM [13,14,34,35]. When SMEs adopt the use of SMM, it is well understood that they will incur lesser costs as compared to other means of marketing. The SMM mechanisms will reach a wider customer base of the SMEs [36–38]. The studies that were cited contend that perceived usefulness, perceived ease of use (TAM model), and compatibility (IDT model) would significantly (positively or negatively) impact on the SMEs to adopt SMM [13]. The studies by Wang, Pauleen, and Zhang (2016) [39] and Chatterjee, et al. (2021) [34] have indicated that the use of SMM mechanisms by SMEs has a significantly positive effect on the enhancement of SME businesses.

## 2.2. The Moderating Effect of Demographics

In order to explain adoption and acceptance of different forms of technological innovations, various researchers have attempted to make modifications on traditional models that impact technology adoption and acceptance [40]. In connection with some of the constructs of the two traditional models (TAM and IDT) that were adopted for this study, also employed are two demographic variables, that is, age and gender level, to investigate their moderating effect between SMM and SMEs' performance. While SMEs' adoption of SMM has been studied extensively, little research does exist which examines the factors that impact on adoption of SMM by SMEs in the Ugandan perspective. Additionally, the moderating influence of demographic variables on SMEs to adopt SMM has not been documented from the reviewed literature. The current work attempts to investigate the moderating effect of two demographic variables, namely age and gender. It is assumed that by incorporating these two moderator variables in the model, the study will minimize the inconsistences that are prevalent in the previous research.

## 2.3. Development of Hypotheses and Conceptual Model

From previous studies [41], and [13] contend that the application of SMM by SMEs would significantly impact on their business. They further state that significant beliefs, namely perceived usefulness, perceived ease of use, and compatibility do exist and could justify SMEs adoption of SMM. These factors together with the demographics that were discussed in Section 3.2 are separately presented to advance the hypotheses and propose a conceptual model.

### 2.3.1. Perceived Ease of Use (PEOU)

PEOU is regarded as the level at which a technology is perceived, easy to understand and operate [40,42]. This belief is also well presented in the Technology Acceptance Model (TAM) [18,43,44]. When an innovation or technology is perceived by the intended users that it is not complicated and can be used with less effort, they will take no time to use the innovation or technology that is brought before them given that it does ease their work operations [45,46]. A relationship with an idea that it would be important to put a concerted effort from the intended user of the technology to be able to use it does exist with this belief [13,47].

There are other components that are contained in this belief, namely self-efficacy [48]. It is perceived that self-efficacy as well impacts on SMEs to adopt and use social media [13]. There is always great motivation from the intended users of the technology if they perceive that the innovation is easier to use [49]. From the reviewed studies, there is relative proof that indeed PEOU has a positively significant relationship with the use of an innovation or new technology. Therefore, in the case that Ugandan SMEs perceive that the application of SMM is indeed easier to use, the proprietors of these SMEs will readily adopt and use SMM. This leads to the formulation of the following hypothesis.

**H1.** *Perceived ease of use (PEOU) has a significantly positive influence on the SMEs to adopt SMM.*

### 2.3.2. Perceived Usefulness (PEU)

Again, the PEU belief is also discussed in the Technology Acceptance Model [18,43]. It is the degree to which one believes that using a specified innovation or technology would improve his or her performance on the job [18]. Thus, this belief evaluates the useful component of an individual adopting an innovation on the basis that it will help them improve or accomplish tasks that are established [50]. In case the proprietors of the SMEs get a feeling that the use of an innovation, that is, SMM, is capable of improving their business efficiency, they will embrace it and use the technology [51]. Chatterjee and Kar (2020) [13] pointed out that a number of studies indicate that PEU has substantial links with the intention of the potential users to use the innovation, for instance, in the

application of a new technology. It is also stated by Syaifullah, et al. (2021) [48] that the PEU belief has a significantly positive association with the ultimate use of the new innovation. Studies have also revealed the same for the application of social media via the smartphone environment. PEU has a significantly positive association towards the application of smartphone technology [52]. PEU is an integral component of various factors, namely performance, effectiveness, as well as risk and trust [53]. Substantial gains can be achieved when risk factors such as sense of privacy and security are highly protected in the use of SMM in SMEs [54]. From the discussed points, an appropriate hypothesis is stated as:

**H2.** *Perceived usefulness (PEU) has a significantly positive influence on the SMEs to adopt SMM.*

### 2.3.3. Compatibility (COM)

Compatibility is closely related to the conception that is affiliated with the extent to which the new technology seamlessly aligns with previous traditional practices and present needs together with the existing values of the SMEs [13]. The reviewed literature showed that the level of compatibility that is presented between the traditional and new innovation products is perceived as effective under substantial scrutiny of users in line with the service [55]. The belief of compatibility as discussed in the Innovation Diffusion Theory (IDT) that was proposed by [27], is considered as an important component for the adoption of innovative technology, which in this study, are SMM technologies. In the case of SMEs, if the perception and relevance to adopt innovation (i.e., the adoption of SMM) mechanisms are compatible with the organizational work application, the SMEs will certainly adopt the innovation [56,57].

Ref. [14] stated that incorporating SMM in SMEs is regarded a best-fit phenomenon. This mechanism is assumed to reach out entirely to probable consumers in the best way possible as well as enhancing the entrepreneur capabilities of firms. From the foregoing submission by [14], there does exist a relative level of augmentation with processes and errands of the business unit such that the businesses that are able to apply SMM seamlessly. From this discussion, the following hypothesis is suggested

**H3.** *Compatibility (COM) has a significantly positive influence on the SMEs to use SMM.*

### 2.3.4. Social Media Marketing (SMM)

There is a significant amount of digital media marketing that seems to be changing at a pace that was not earlier anticipated [58]. Information technology, in principal the World Wide Web, has shown a positive impact on the entrepreneurial industry over the past decade (Wu, 2006) [59]. A definition that was coined by [60] and [61] is that it is 'a secured generation of web development and design that aims to facilitate communication, sources Wide Web'. It is estimated that product consumers are used to spending about 6 hours daily with involvement in the social media environment. Online communication between business units and consumers has readily become very easy due to the presence of the social media environment in our reach and in real-time, and it is quickly growing in Ugandan SMEs; this is largely attributed to resource limitations [62].

When enterprises embrace social media usage, there is a high likelihood that their business activities will gain tremendous improvement as they build up their brand to a wider consumer base [13]. It is also anticipated that SMM will assist the SMEs in Uganda to invest more in technological marketing mechanisms. From the submissions, we thus formulate the following hypothesis:

**H4.** *Social media marketing (SMM) adoption has a significant positive effect on the performance of SMEs in Uganda.*

*2.4. Hypotheses Development for the Moderating Variables*

Moderating variables can be loosely defined as those variables that can, in any direction (strengthen, reduce, and negate), influence the predictor and predicted variables [63]. The moderating variables that are considered in this study are age and gender. It is thus important to investigate how these influence the SMEs in adopting SMM in Uganda. Their hypotheses are formulated as indicated below.

### 2.4.1. The Moderating Effect of Age

Research that is related to innovation uptake by various scholars indicates that younger users show a different attraction and behavior as compared to their older counterparts. The latter individuals are said to be somewhat laid back in terms of adopting and using new innovation technologies for various reasons, such as security, compatibility, and complexity. They rely more on what they are fond of, say face to face daily transactions [40]. Research that was conducted by [64] indicated that age has a significant impact on new product ownership in the consumer electronics category. (Again, 2005) [65] documented that age has a moderating effect between technology use and perceptions.

**H5.** *The influence of social media marketing (SMM) on performance of SMEs when moderated by age will improve the performance of SMEs in Uganda.*

### 2.4.2. The Moderating Effect of Gender

The world health organization (WHO) posits that the term gender is employed to characterize the attributes of both women and men that are socially constructed. Various factors that impact technology adoption and acceptance have been proposed by a number of researchers. Ref. [66] did assess the effect of gender as a moderating factor on consumer intention to switch from traditional to online groceries, and the results indicated that gender significantly impacts the factors affecting consumer intention to switch. Ref. [40] studied the moderating effect of gender on mobile banking adoption and found out that gender has a moderating effect between the ease of use and attitude towards mobile banking. The effect was stronger for the women than their male counter parts. Thus, the following hypothesis is developed:

**H6.** *The influence of Social Media Marketing (SMM) on performance of SMEs when moderated by gender will improve on the performance of SMEs in Uganda.*

## 3. Research Methodology

SMEs performing their entrepreneurial activities in Kabale district, Uganda, were primarily targeted. On the basis of this, a structured survey was preferred and, therefore, adopted. A total of 152 SMEs (observation units) and their proprietors (unit of analysis) were selected via cluster random sampling. Observing the conceptual model, Figure 1, it is clear that the explanatory variables outnumber the explained variables. The hypotheses were tested using structural equation modelling (SEM) through SPSS version 23.0 and ONYX version 1.14.32, graphical open source software. With SEM being a second generation approach of performing multivariate analysis [40,67], it was the more preferred method in this research.

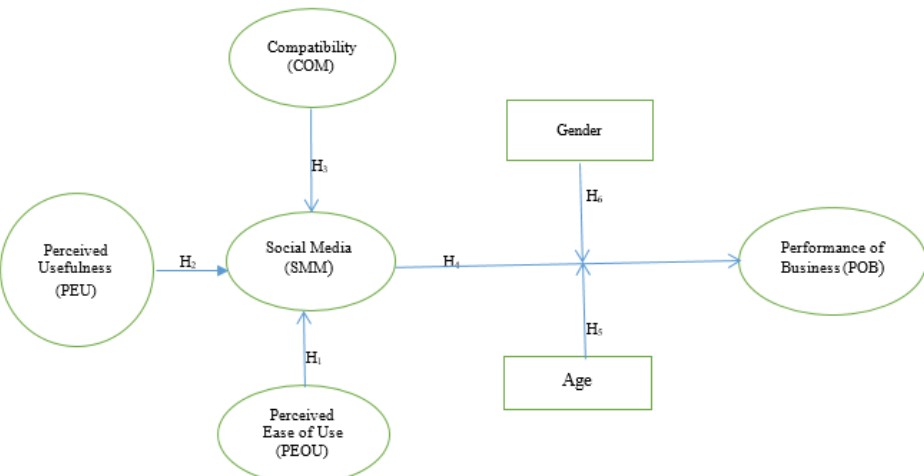

**Figure 1.** Conceptual model.

### 3.1. Instrumentation

When preparing a questionnaire for a specified task, it is important to put focus on the wording [13]. For this study, this was given attention by adapting and customizing items from the scale that was developed by [13]. Before the actual survey is conducted, it is of essence to pre-test the research instruments. This is known as a pilot test for the survey instruments [68]. A total of 10% of the study population is considered for the pilot study. In this work, a sample of 15 was considered for pre-testing. Pre-testing the survey instrument before the actual study helps in minimizing the flaws that could be present in the questionnaire [68] and the final questionnaire is prepared thereafter. The questionnaire had a total of 18 items. The experts that were consulted in this study were the business managers, owners, or proprietors. The research instrument contained closed-ended questions in the form accounts that were associated to a 5-point Likert Scale (SA—strongly agree = 1, A—agree = 2, N—not sure = 3, D—disagree = 4 and SD—strongly disagree = 5). Appropriate data responses were collected for subsequent analysis after thoroughly numerating the feedback.

### 3.2. Data Collection Strategy

Validation of the developed hypotheses and the conceptual model, Figure 1, was done using the PLS-SEM analysis procedure. We have proceeded to validate the hypotheses and the conceptual model with the help of the PLS-SEM analysis technique. However, this study having considered a sample size of 152 respondents (SME owners or managers), did not deter the researchers from proceeding with the study, even though some studies state that an inadequate sample size leads to decreased generalization of the study findings [69]. Usually, a sample size of 300 is highly recommended [70]. On another hand, [71] recommends that item response should lie between 1:4 to 1:10. In this study, the item number is 18, thus, it is prudent that responses lie between 72–180. A total of 152 business units (SMEs) were subsequently considered in this study particularly from Kabale Municipal Council, Kabale District-Uganda, East-Africa. The data were collected through a self-administered questionnaire that was distributed by a research assistant in the months of November and December 2021. Quality control in this study was achieved through a number of steps including a pilot study, thorough variable labeling, and data entry validation guidelines.

A multistage cluster sampling procedure was adopted and the population of interest (SMEs classified according to their capital basis) was clearly identified. Using already-existing units (wards), the municipality was grouped. Kabale Municipality consists of eight (8) wards and each ward was taken into account in order to reach the full target demographic (SMEs), and we made sure there was no overlap. Using a random number table, simple random sampling was utilized to choose the necessary four (4) wards. We chose to randomly select a sample of SMEs from inside the specified wards rather than

collecting data from every SME there. Since there are various numbers of SMEs in each ward, proportionate stratified sampling [72] was used to collect the right sample from each ward. Cluster sampling being a probability sampling method in which the target population is divided into a number of subgroups or clusters for study [73]. In this work, the clustering variable was the estimated capital of the enterprise. This information was readily available from the sampling frame where businesses are classified as small or medium depending on their capital base and number of employees.

### 3.3. Test for the Moderation Effect of the Variables

Testing for the moderating effect of age and gender, a multiple regression model that was suggested by Sharma et al. (1981) was employed, Equation (1):

$$Y = \beta_0 + \beta_1\xi_1 + \beta_2\xi_2 + \beta_3(\xi_1\xi_2) + \varepsilon \tag{1}$$

where $Y$ = categorical variable (performance); $\xi_1$ is the independent variable and $\xi_2$ = categorical moderator variables (age i.e., $\xi_1 = 1$ for age range 18–23, $\xi_1 = 2$ for age range 24–29, $\xi_1 = 3$ for age range 30–35, and $\xi_1 = 4$ for age range 36–41; $\xi_2$ dichotomous moderator variable gender. With a value of 1 and 0 (e.g., $\xi_2 = 1$ for female and $\xi_2 = 0$ for male) and $\xi_1\xi_2$ = the interaction term between the explanatory and moderator variables. $\varepsilon$ is the stochastic error term.

To have $\xi_2$ as a moderator variable, $\beta_3$ has to be statistically significant, while $\beta_1$ and $\beta_2$ should be statistically insignificant [40].

## 4. Data Analysis and Results

### 4.1. Descriptive Statistics

Table 1 shows the background characteristics of the entrepreneurs for the considered sample. More males, 92 (60.53%) than females 60 (39.47%) participated in the survey, with the majority of the entrepreneurs aged between 30 and 35 years, that is 22 (36.1%) and 39 (42.9%) for both the female and male categories, respectively. Further in Table 1, O-level refers to the ordinary level of the certificate of education, and A-level is the advanced level of the certificate of education. From both genders, degree holders were the majority operators of the business units, with 37.1% females and 50.5% males. In terms of the number of years spent in business or experience, females who had spent between 7 and 10 years were 21.0% while their males counterparts with the same experience were 26.4%. It can be concluded that there was no significant difference in the experience of operating businesses among the gender categories.

**Table 1.** Background characteristics of the entrepreneurs.

| Gender. | Categories | #(%) | | Years # (%) |
|---|---|---|---|---|
| *Female* | | | | |
| ● Age | 18–23 | 6(9.8) | ● Experience | 1–3 22(35.0) |
| | 24–29 | 18(29.5) | | 4–6 19(30.6) |
| | 30–35 | 22(36.1) | | 7–10 13(21.0) |
| | 36–41 | 15(24.6) | | >11 8(12.9) |
| ● Educ_level | O_level | 8(1.9) | | |
| | A_level | 10(16.1) | | |
| | Diploma | 20(32.3) | | |
| | Degree | 23(37.1) | | |
| | Others | 1(1.6) | | |
| *Male* | | | | |

**Table 1.** *Cont.*

| | Gender. | Categories | #(%) | | | Years # (%) |
|---|---|---|---|---|---|---|
| • | Age | 18–23 | 6(6.6) | • | Experience | 1–3 23(25.3) |
| | | 24–29 | 31(34.1) | | | 4–6 28(30.8) |
| | | 30–35 | 39(42.9) | | | 7–10 24(26.4) |
| | | 36–41 | 15(16.5) | | | >11 16(17.6) |
| • | Educ_level | O_level | 9(9.9) | | | |
| | | A_level | 14(15.4) | | | |
| | | Diploma | 20(22.0) | | | |
| | | Degree | 46(50.5) | | | |
| | | Others | 2(2.2) | | | |

Exploratory Factor Analysis (EFA)

It is important to note that factor analysis has assumptions that have to be put into consideration. (i) The sample size. Mundfrom, Shaw, and Ke (2005) [74] stated that a number of suggestions are available for the sample size to be used in conducting a factor analysis. They further suggest that a minimum sample size of 3–20 times the number of variables as well as absolute ranges from 100 to above 1000. (ii) Assumes reliable correlations, this is highly put into consideration in the presence of huge missing data. In this study, this will be looked at using the Kaiser–Meyer–Olkin and Bartlett's measure of sampling adequacy (KMO and Bartlett's test). (iii) Normality. This is not an assumption of EFA so it is not an issue in this analysis. Finally, (iv) multicollinearity/singularity. For this, the determinant of the data matrix will be used to test if it is a problem or not.

*4.2. Analysis of the Measurement Model*

The measurement model of any study requires the ability to achieve reliability, validity, and uni-dimensionality [75,76]. Uni-dimensionality is achieved if the factor loadings of all the items are positive, which is the case for this study (Table 2). Again, seven constructs have been considered, see conceptual model, Figure 1. The lower limit value of 0.6 of the Cronbach's alpha has been considered for the study [77]. Table 2 shows the results of the measurement model. It is observed that the lowest value of Cronbach's alpha that was estimated is greater than 0.6. Thus, it is concluded that all the constructs that were considered in this study are reliable and consistent [13].

**Table 2.** Validity and reliability: model measurements.

| Construct/Item | Loading | Cronbach's Alpha | Composite Reliability (CR) > 0.6 | Average Variance Extracted (AVE) > 0.5 |
|---|---|---|---|---|
| Compatibility (COM) | | 0.754 | 0.625 | 0.692 |
| COM1 | 0.658 | | | |
| COM2 | 0.556 | | | |
| COM3 | 0.663 | | | |
| COM4 | 0.596 | | | |
| COM5 | 0.629 | | | |
| Perceived usefulness (PEU) | | 0.654 | 0.619 | 0.722 |
| PEU1 | 0.590 | | | |
| PEU2 | 0.579 | | | |
| PEU3 | 0.463 | | | |
| PEU4 | 0.445 | | | |

**Table 2.** *Cont.*

| Construct/Item | Loading | Cronbach's Alpha | Composite Reliability (CR) > 0.6 | Average Variance Extracted (AVE) > 0.5 |
|---|---|---|---|---|
| Perceived ease of use (PEOU) | | 0.750 | 0.781 | 0.678 |
| PEOU1 | 0.559 | | | |
| PEOU2 | 0.858 | | | |
| PEOU3 | 0.919 | | | |
| PEOU4 | 0.956 | | | |
| PEOU5 | 0.935 | | | |
| Performance of business (POB) | | 0.682 | 0.845 | 0.775 |
| POB1 | 0.322 | | | |
| POB2 | 0.858 | | | |
| POB3 | 0.944 | | | |
| POB4 | 0.901 | | | |
| POB5 | 0.967 | | | |
| Social media marketing (SMM) | | 0.723 | 0.802 | 0.765 |
| SMM1 | 0.78 | | | |
| SMM2 | 0.75 | | | |
| SMM3 | 0.77 | | | |

The loadings for COM1 are interpreted as 65.8% of the variance in compatibility can be alluded to Question 1 (COM1), and so on. In the loadings column, loadings that are less than 0.3 are the major focus and could indicate that there is a problem with that particular question as it may not be in conformity with other questions in trying to measure the same concept. For this study, all the questions do not seem to have a problem as they have loadings above 0.3. The composite reliability (CR) in Table 2 is also computed to ascertain the internal consistency in scale items [78]. The threshold values for the CR are differently suggested, with some authors stating that an appropriate threshold could be from 0.60 and above [79]). In computing the average variance extracted (AVE) for the constructs, the interest is to know how much variation in the considered items can be explained by the latent variable. From Table 2, for instance, the latent variable or construct compatibility (COM) was measured with five items and the AVE for these items is 0.692, implying that, on average, 69.2% of the variation in SMM's compatibility in SMEs is explained by these five items or questions.

*4.3. Discriminant Validity*

The principal reason for carrying out discriminant validity in this study is to reveal how distinct an item or set of items (research questions) is from others [80]. The interest in this section is to show that the research questions measuring the various latent variables have poor association or low correlation existing among them. This analysis is important as there could be a likelihood that some questions will possess a significant association with questions for which they are not intended to measure the same latent variable [81–83].

In this study, the Fronell–Larcker Criterion [84] is employed along with the heterotrait-monotrait (HTMT) ratio of correlation to test for discriminant validity of the measurement model. As per this criterion, the condition for achieving discriminant validity is that the square root of the average variance that is extracted by a latent variable/construct should be greater than the correlation between the latent variable and any other latent variables. The averages (COM_AV, PEU_AV, PEOU_AV, and POB_AV) for each latent variable were computed using an appropriate procedure in SPSS. However, the main interest here is in the values of the Pearson correlation together with the asterisk (*) which indicates the significance level, Table 3. For the HTMT criterion [85], it is able to obtain greater specificity

and sensitivity rates (97% to 99%) compared to the cross-loadings criterion (0.00%) and Fornell–Lacker (20.82%), according to a Monte Carlo simulation research that was used to demonstrate the method's superior performance Table 4.

**Table 3.** Discriminant validity test using the Fornell–Lacker criterion.

| Latent Variable | COM_AV | PEU_AV | PEOU_AV | POB_AV | AVE |
|---|---|---|---|---|---|
| COM_AV | 0.882 | | | | 0.692 |
| PEU_AV | 0.624 ** | 0.850 | | | 0.722 |
| PEOU_AV | 0.636 ** | 0.742 ** | 0.823 | | 0.678 |
| POB_AV | 0.596 ** | 0.668 ** | 0.929 ** | 0.880 | 0.775 |

** Correlation is significant at the 0.01 level (2-tailed).

**Table 4.** Discriminant validity test using heterotrait-monotrait (HTMT) criterion.

| Latent Variable | COM | PEU | PEOU | POB |
|---|---|---|---|---|
| COM | - | | | |
| PEU | **0.94** | - | | |
| PEOU | 0.70 | **0.92** | - | |
| POB | 0.78 | **0.91** | 0.82 | |

From Table 3, it can clearly be seen that the correlations of a latent variable with other latent variables are less than the square root of its AVE. It is then noticed that the criterion is satisfied, thus discriminant validity has been successfully established for this research.

Table 4 indicates the results from HTMT assessment. The formula from Henseler et al. (2015) [85] can be used to calculate this outcome from HTMT.

From the HTMT results, Table 4, figures in bold indicate an issue with the discriminant validity according to the HTMT 0.90 criterion. Stating that the latent constructs' collinearity/multicollinearity issues are detected by the HTMT criterion. It should be highlighted that the discriminant validity has been demonstrated between two reflective constructs if the HTMT value is less than 0.90 [86]. The constructs of compatibility-perceived usefulness, perceived usefulness-perceived ease of use, and perceived usefulness-performance of business are revealing discriminant issues/problems. In this case, it could be that the majority of the items are measuring the same thing. As a result, it contains the overlapping items from the participants' views in the affected constructs. However, the Fornell–Lacker analyses findings were mainly taken into account by the study's researchers for additional evaluations and modelling of the necessary framework for the issue at hand.

### 4.4. PLS-SEM Analysis

This is an approach for structural equation modelling that enables one to estimate multifaceted cause-effect relationships in path models that possess latent variables [87]. Thus, PLS-SEM [88] is employed in this study to evaluate the proposed conceptual model, Figure 1, by fitting different indices such as Chi-square, RMSEA, CFI, and RMSR, indicated in Table 5. In structural equation models (SEM), the threshold values, Table 5, are taken as the minimum values for the computed indices such as in [89].

**Table 5.** Model fit indices.

| Fit Indices | Threshold Values | Model Values Obtained |
|---|---|---|
| Chi-square ($\chi^2$) | Its *p*-value should be >0.05 | 0.064 |
| Root mean square error of approximation (RMSEA) | ≤0.05 | 0.002 |
| Comparative fit index (CFI) | >0.90 (Fan et al., 1999) | 0.936 |
| Root mean square residual (RMSR) | <0.06 or 0.08 | 0.027 |

From Table 5, all the estimates for the various indices are within tolerable ranges. Thus, the model is a good fit. It should be recalled that for model estimation, a graphical interface software ONYX was used.

From the empirical research, the estimated model is indicated in Figures 2 and 3 with exhaustive estimates shown in Table 6.

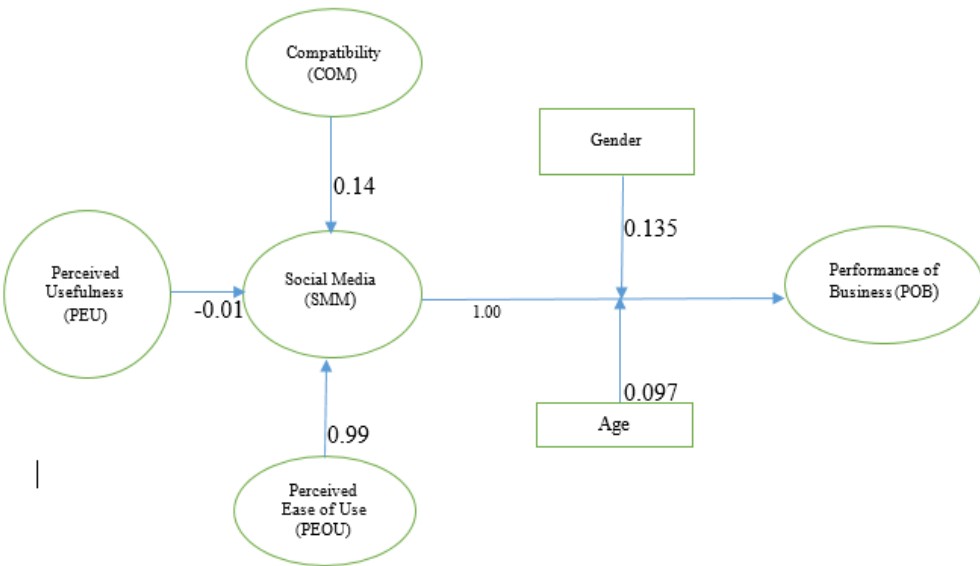

**Figure 2.** Structural model with path weight.

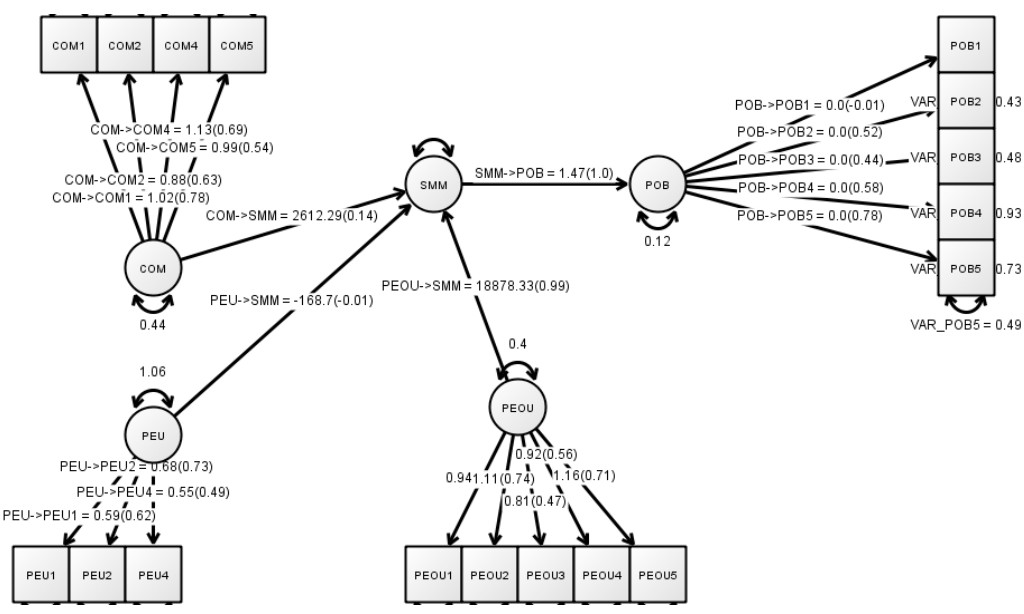

**Figure 3.** Estimated structural model summarized in Figure 2.

**Table 6.** Results of hypotheses testing.

| Path | Hypothesis | Path Coefficient | *p*-Value | Remark |
|---|---|---|---|---|
| PEOU → SMM | H1 | 0.99 | * ($p < 0.05$) | supported |
| PEU → SMM | H2 | −0.01 | ** ($p < 0.01$) | supported |
| COM → SMM | H3 | 0.14 | ** ($p < 0.01$) | supported |
| SMM → POB | H4 | 1.00 | ** ($p < 0.01$) | supported |

### 4.5. Moderation Analysis: SPSS

In order to ascertain whether the relationship between two variables depends on a third variable, it is important to perform a moderation analysis [90]. Having categorized the variable age in this study and centralized all other variables, this relationship is between a continuous dependent variable and a continuous independent variable. The standardized approach of establishing whether a moderating influence does exist involves the integration of an interaction term in a multiple regression model [90]. A moderation analysis is practically a multiple linear regression approach involving an interaction term [91]. However, before moderation analysis is performed, it is advisable that the assumptions on the data that are prepared for analysis should be met [92].

On running a reference case, ANOVA analysis involving only the independent variables PEOU_AV, COM_AV, PEU-AV (the centralized values of the initial variables as earlier stated), and the dependent variable POB_AV should be run. It is noticed that these variables have a significant impact ($p < 0.05$) on the performance of the business unit (POB_AV), Table 7.

**Table 7.** ANOVA results for the regression analysis.

| Model | | Sum of Squares | df | Mean Square | F | Sig. |
|---|---|---|---|---|---|---|
| | Regression | 45.210 | 3 | 15.070 | 256.639 | 0.000 [b] |
| 1 | Residual | 8.749 | 149 | 0.059 | | |
| | Total | 53.960 | 152 | | | |

Dependent Variable: POB_AV; [b]. Predictors: (Constant), PEOU_AV, COM_AV, PEU_AV.

At this stage, another independent variable, gender (moderating) is introduced into the model and this could be a rival to the independent variables in influencing the outcome, POB_AV. On the first run, when the gender variable is introduced into the model, it appears that gender does not have a moderating influence. However, on introducing the interaction variable interaction_gender, a moderation effect is noticed as the new variable has a significant effect ($p < 0.05$) on POB_AV, Table 8.

**Table 8.** Regression results with gender as a moderator variable.

| Model | | Unstandardized Coefficients | | Standardized Coefficients | t | Sig. | Collinearity Statistics | |
|---|---|---|---|---|---|---|---|---|
| | | B | Std. Error | Beta | | | Tolerance | VIF |
| | (Constant) | 0.276 | 0.096 | | 2.884 | 0.005 | | |
| | PEOU_AV | 0.774 | 0.042 | 0.951 | 18.303 | 0.000 | 0.392 | 2.553 |
| 1 | GENDER | 0.068 | 0.050 | 0.056 | 1.364 | 0.175 | 0.620 | 1.614 |
| | COM_AV | 0.053 | 0.038 | 0.064 | 1.403 | 0.163 | 0.504 | 1.984 |
| | PEU_AV | −0.001 | 0.043 | −0.001 | −0.015 | 0.988 | 0.382 | 2.620 |
| | Interaction_gender | −0.007 | 0.003 | −0.135 | −2.484 | 0.014 | 0.357 | 2.799 |

Dependent Variable: POB_AV.

The regression results with gender (with only two categories) as a moderator variable are presented in Table 8. The results show that gender does moderate the relationship between each of the independent variables, namely, compatibility, perceived usefulness, and the perceived ease of use, with the dependent variable performance of business (SMEs). This is because the interaction effect (interaction_gender) for the combination of these variables is statistically significant ($p < 0.05$). It can be concluded that the gender of the business proprietor has a significant effect on the performance of the business.

Hence, H6 is supported as seen in Table 8, with $t = |2.484| > 1.96$ and $p$-value = 0.01 which is less than the significance level of 0.05.

The second moderator variable that was considered was age which had four (4) categories, namely "18–23", "24–29", "30–35", and "36–41". As a result, three dummy variables for the age variable were created and their respective interaction terms were introduced into the regression models as indicated in Tables 9–11. The results from Table 9 indicate that age moderates the relationship between social media marketing (SMM) and the performance of business (POB). It is observed that for business proprietors in the age category "18–23", the influence of SMM on POB is statistically significant ($p < 0.1$). Again, it is assumed that the effect of SMM on POB is higher for proprietors in the age category "18–23" as compared to other age categories. Tables 10 and 11 indicate that age does not moderate the relationship between SMM and POB. In both tables, the impact of either age categories is not statistically significant ($p > 0.1$). A conceivable reason for this could be that elderly proprietors have a low intake/ adoption for technology innovations as compared to their younger counterparts in the lower age categories.

**Table 9.** Regression results with the age "18–23" category as a moderator variable.

| Model | | Unstandardized Coefficients | | Standardized Coefficients | t | Sig. | Collinearity Statistics | |
|---|---|---|---|---|---|---|---|---|
| | | B | Std. Error | Beta | | | Tolerance | VIF |
| 1 | (Constant) | 0.438 | 0.069 | | 6.357 | 0.000 | | |
| | COM_AV | 0.030 | 0.037 | 0.036 | 0.798 | 0.426 | 0.537 | 1.863 |
| | PEU_AV | −0.046 | 0.042 | −0.055 | −1.080 | 0.282 | 0.407 | 2.455 |
| | PEOU_AV | 0.758 | 0.042 | 0.932 | 18.259 | 0.000 | 0.410 | 2.439 |
| | AGECAT1 | −0.074 | 0.121 | −0.033 | −0.610 | 0.543 | 0.354 | 2.825 |
| | Interaction_Agecat1 | 0.027 | 0.015 | 0.097 | 1.768 | 0.079 | 0.358 | 2.795 |

Dependent variable: POB_AV.

**Table 10.** Regression results with the age "24–29" category as a moderator variable.

| Model | | Unstandardized Coefficients | | Standardized Coefficients | t | Sig. | Collinearity Statistics | |
|---|---|---|---|---|---|---|---|---|
| | | B | Std. Error | Beta | | | Tolerance | VIF |
| 1 | (Constant) | 0.472 | 0.078 | | 6.018 | 0.000 | | |
| | COM_AV | 0.023 | 0.038 | 0.028 | 0.623 | 0.534 | 0.533 | 1.875 |
| | PEU_AV | −0.039 | 0.043 | −0.047 | −0.914 | 0.362 | 0.406 | 2.462 |
| | PEOU_AV | 0.747 | 0.042 | 0.917 | 17.694 | 0.000 | 0.408 | 2.453 |
| | AGECAT2 | −0.060 | 0.068 | −0.047 | −0.894 | 0.373 | 0.389 | 2.573 |
| | Interaction_Agecat2 | 0.006 | 0.006 | 0.052 | 0.989 | 0.324 | 0.395 | 2.534 |

Dependent variable: POB_AV.

**Table 11.** Regression results with the age "30–35" category as a moderator variable.

| Model | | Unstandardized Coefficients | | Standardized Coefficients | t | Sig. | Collinearity Statistics | |
|---|---|---|---|---|---|---|---|---|
| | | B | Std. Error | Beta | | | Tolerance | VIF |
| 1 | (Constant) | 0.411 | 0.086 | | 4.786 | 0.000 | | |
| | COM_AV | 0.031 | 0.038 | 0.037 | 0.807 | 0.421 | 0.529 | 1.889 |
| | PEU_AV | −0.030 | 0.043 | −0.036 | −0.687 | 0.493 | 0.402 | 2.487 |
| | PEOU_AV | 0.762 | 0.044 | 0.936 | 17.167 | 0.000 | 0.369 | 2.708 |
| | AGECAT3 | −0.002 | 0.051 | −0.002 | −0.041 | 0.967 | 0.618 | 1.619 |
| | Interaction_Agecat3 | −0.002 | 0.003 | −0.032 | −0.617 | 0.538 | 0.419 | 2.389 |

Dependent variable: POB_AV.

Thus, H5 is supported under the age category "18–23".

## 5. Summary of Results

The theorized factors in this work, that is, the perceived ease of use (PEOU), perceived usefulness (PEU), and compatibility (com) were all derived from the conception of TAM [18,43] and Innovation Diffusion Theory (IDT) [18]. All the hypotheses (H1–H4) were

formulated from the conceptualization of work that was done by [18] and [27]. Modification of the IDT framework was enhanced by adding two moderating variables (age and gender) in the model that was proposed by [27] Rogers 1983. The established hypotheses PEOU → SMM (H1), PEU → SMM (H2), COM → SMM (H3), and SMM → POB (H4) were found to be supported as a result of the data analysis. The hypotheses that were formulated from the moderating variables, that is H5 on age and H6 on gender, were found to be supported. However, for age only the age category "18–23" was found to have an impact on the performance of businesses after the application of SMM. The relationships of these moderating variables to PEOU, PEU, and COM were validated via the statistical analyses that were performed and have also gained support from previous work [40].

## 6. Discussions of Results

The PEOU → SMM (H1) and PEU → SMM (H2) theories were developed from the conception of TAM [18]. It is discovered through the current empirical study that these two theories have been supported. This is in line with earlier research [47] that investigated how PEOU and PEU affected SMM using TAM. Additionally, because PEU incorporates these crucial ideas, the impact of PEU on SMM (H1) denotes that it also considers the simultaneous influences of performance, effectiveness, risks, and trust on SMM [47]. Additionally, a second exogenous variable, PEOU, contains components such as self-efficacy and technological simplicity [47]. Since these two beliefs have an impact on SMM, PEOU's impact on SMM (H2) also takes that into account. Once more, the idea for the theorized COM→SMM (H3) originated with IDT [27]. COM has been found to have a positive and significant impact on social media marketing in the current work (H3). This connection has been verified statistically, and earlier research has provided additional support [13]. Furthermore, this study analyzed the impact of age and gender as moderating variables. A moderating variable is a factor that can improve, worsen, negate, or otherwise change the relationship between independent and dependent variables. In this study, we sought to determine how these factors affected the decision of the owners of SMEs to implement SMM. They were hypothesized as SMM (age) → POB (H5) and SMM (gender) → POB (H6). According to statistical study, both gender and age were shown to positively affect the operation of the firm, albeit only the age range of "18–23" was found to be significant. Here, it can be deduced that young people in Uganda who are between the ages of "18 and 23" frequently utilize social media to advertise their enterprises. SMM has also been proven to have an effect on POB when moderated by gender. This suggests that whether a business unit adopts SMM or not will depend on the gender of the business owner and how that would affect the performance of the SME particularly in the Ugandan setting.

### 6.1. Theoretical Contributions

In this work, three components PEU, PEOU, and COM have been hypothesized and moderated by demographics such as age and gender to encourage SMEs to adopt SMM for improved business performance (POB). When SMM adoption by SMEs is moderated by owner age and gender, their performance is improved. The improvement is referred to as performance of business unit (POB) in this context, and it is the study's main goal. The current analysis has been able to determine whether SMM use by Ugandan SMEs is required in order to increase their commercial production. When including perceived usefulness, a variety of factors were taken into account, including performance (of SMM), trust in SMM, risk perception when using SMM, effectiveness when using SMM, and productivity [13]. The perceived ease of use (PEOU), the second factor, also involves self-efficacy and simplicity [13]. In this instance, it is obvious that a number of factors that could encourage SMEs to adopt SMM have been considered as a result of including the two aforementioned components, PEU and PEOU. The information that was provided could be viewed as the study's theoretical contribution.

### *6.2. Practical Implications*

Our knowledge of the applicability of the TAM and IDT theories as well as the significance of age and gender factors for improved SME performance has been improved by this study. Additionally, this study is an expansion of past research on how PEOU, PEU, and COM theories affect SME performance after implementing SMM. The link between SMM and SME performance and the moderating effects of age and gender has been established. This study also showed that adopting SMM by SMEs to enhance the performance of the business unit is significantly influenced by age and gender factors in business ownership. As a result, this study might serve as a point of reference for academics in the entrepreneurship and marketing fields as they try to figure out what other empirical linkages they might make with regard to SME performance and sustainability. Although the current study has merit, there are some drawbacks as well.

### 7. Limitations and Directions for Future Research

In the development of the model that is presented in Figure 2, care was taken to follow the previous studies that were done by researchers such as [18] and [27]. However, limitations cannot be ruled out in the current work. In this study, entrepreneurs were selected from Kabale district. Uganda has several districts, and inputs of entrepreneurs from other districts were not taken into account. Consideration of views from other SMEs in Uganda could have probably given different results. In that perspective, the results of this study cannot be considered representative of SMEs in the entire Uganda. Further research is needed to expand the scope of this work and obtain a representative sample from Uganda SMEs such that the results can be regarded as generic.

### 8. Conclusions

There were two known theories, TAM and IDT, that were used to assess the influence of the variables such as perceived ease of use, perceived usefulness, compatibility, gender, and age on the performance of businesses towards the adoption of social media marketing. The study proposes a new model by incorporating two additional background variables of the business proprietor which have not been given attention from the reviewed literature. In this work, the moderating effect of age and gender on SMM toward POB was examined using the regression analysis.

The regression analysis results in Tables 8–11 indicate that the explanatory variable (s) for "interaction_gender" and age group "18–23" have a statistically significant impact on SMM toward POB as indicated by the studentized t-statistic.

In this study, focus has been put on how various factors could possibly enhance the SMEs in Uganda and also establish an all embracing environment to adopt social media marketing techniques so as to realize an improved business performance. The results of this work are anticipated to give thorough and informative knowledge to business proprietors and policy-makers to rethink the current policies and put in place relevant policies that cope with the present digital environment.

**Author Contributions:** Conceptualization, C.R.K., C.K. and M.A.; Formal analysis, C.R.K., D.N., A.S. and B.M.K.; Funding acquisition, B.M.K. and C.K.; Investigation, C.R.K., M.A., D.N. and B.M.K.; Methodology, C.R.K. and B.M.K. and A.S.; Project administration, C.R.K. and B.M.K.; Supervision, C.R.K. All authors have read and agreed to the published version of the manuscript.

**Funding:** This research received no funding from any organization.

**Informed Consent Statement:** Informed consent was obtained from all subjects involved in the study.

**Data Availability Statement:** Data can be obtained from the Directorate of Research and Innovation, Kable University.

**Conflicts of Interest:** The authors declare no conflict of interest.

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
