# Peer review of "Social Media Marketing for Small and Medium Enterprise Performance in Uganda: A Structural Equation Model"

_sustainability, doi:10.3390/su142114391_

Round 1

Reviewer 1 Report

Dear authors,

the article on Social Media Marketing for Small and Medium Enterprises performance in Uganda, proposed for the journal Sustainability is actual and of interest to a broader audience.

It has the appropriate scientific soundness and the model is correctly developed. Stated are also aims, research questions and hypotheses. Provided is also the overview of the literature.

However, the article is missing a discussion after the results of empirical analysis. It is an essential part of each scientific article, as it provides in-depth insight into the topic, together with the comparison of your analysis results to other authors’ results. It also provides readers with a contribution to the science of your research. Therefore, the Discussion part has to be added to the article.

Additionally, the literature is not organized according to the template/instructions of the article preparation. In the text, there are also some typing / technical errors (e.g., bracket in line 156; semicolon in line 157…).

Wish you good luck with the article improvements!

Author Response

I have duly responded to the comments the original version of the paper and attached is the rebuttal.

Reviewer 2 Report

I have finished the review on the manuscript “Social media marketing for small and medium enterprises performance in Uganda: A structural equation model”(Manuscript ID: sustainability-1907992).

This manuscript examined the relationship between the use of social media marketing and the performance of small and medium enterprises in Uganda.

Two theoretical frameworks—technology adoption model (TAM) and the model of diffusion of innovation (IDT)—were adopted to guide this study. As stated in the manuscript, this study employed the method of cluster random sampling to distribute the questionnaires and 152 valid questionnaires were obtained. Three variables that were derived from the two theoretical frameworks were perceived usefulness, perceived ease of use, and compatibility, which were found to be significant predictors for the performances of small and medium enterprises. Furthermore, age and gender were discovered to be moderators of the relationship between the use of social media marketing and the performance of small and medium enterprises.

My comments on this manuscript were as followed:

The issue is interesting and the two theoretical frameworks were appropriate to examine this issue. However, there are several major flaws that need to be fixed in order for this manuscript to be publishable.

(1) In the introduction section, the authors have clearly explained the background of Uganda in terms of the role small and medium enterprises play in this country. Therefore, I suggest that the first three paragraphs in the literature review can be deleted because they are repetitive in terms of the background of Uganda.

(2) The authors did not adopt the whole theoretical frameworks but chose two variables from TAM and one variable from IDT. Therefore, the authors should give reasons why chose the three variables. In fact, Rogers’ model identifies perceived innovation attributes as the most important predictor of technology adoption and innovation attributes contain five variables—relative advantages, compatibility, complexity, trialability, and observability. As indicated in IDT, ease of use is similar to the concept of complexity and usefulness is one of the relative advantages.

(3) Rogers’ model divides all consumers into five types of adopters: innovators (the first 2.5%), early adopters (the next 13.5%), early majority (the next 34%), late majority (the next 34%), and laggards (the last 16%). Empirical studies have shown that the five types of adopters differ significantly in their demographic variables. Furthermore, Rogers’ model suggests that demographics are important variables in the early stages of an innovation’s diffusion and are capable of differentiating adopters from non-adopters. As the diffusion of an innovation comes into its late stages, demographics will no longer exert a significant effect on adoption. Empirical research shows that early adopters are more likely to be male, younger and better educated and to have higher incomes than non-adopters.

Therefore, based on Rogers’ model or TAM, demographic variables do not exert any moderating effects on the adoption. If the authors claimed that demographic variables do have moderating effects on the adoption or on the performances, then the authors should provide strong justifications for this claim.

(4) In the second paragraph on page 5, the authors stated that they would examine the moderating influence of age and educational level. Similarly, they mentioned to examine the moderating influence of age and educational level in the fourth paragraph on page 7. The same statement appeared in the second paragraph on page 9. But in the actual study, H6, and Figure 1, they measured the moderating influence of age and gender.

(5) In the methodology, the authors claimed that they adopted the method of cluster random sampling by using the estimated capital of the enterprise as clusters. However, the authors did not provide the following information including the range of clusters, how many clusters there were, and how many enterprises they chose from each cluster. They also have to indicate how representative of their final sample to the population of small and medium enterprises.

(6) There is a lack of operationalization of major variables in this manuscript. Particularly, the authors need to indicate how the authors measured the variables of social media marketing and performances of small and medium enterprises. Without these data, it would be difficult for readers to estimate the validity of the two major variables.

Author Response

Rebuttal attached

Reviewer 3 Report

Dear authors, it was my pleasure to review your manuscript.

In the following, I send you my observations that reflect the need for some details that would have helped me to understand the research design more easily.

Regarding the abstract:

 - whose age and education? (line 20)

- in what way does the study contribute to the existing literature? (line 30)

- to specify who the respondents were.

The abstract should indicate the innovation of the research.

 Regarding the body of the paper:

- I suggest a clear definition of each variable; for example, Perceived ease of use by...; Perceived usefulness by...

- I also suggest these clarifications in formulating the hypotheses: Perceived ease of use by...; Perceived usefulness by...; Compatibility between...

- Hypothesis 4 should contain `adoption': Social Media Marketing adoption

- Under table 1, I think it is necessary to briefly explain the symbols regarding the level of education.

- Regarding the respondent, you mention in the abstract: `Empirical model validation has been performed using 151 business units` and in the section 4.2: `this study having considered a sample size of 152 respondents… Cluster random sampling was employed in the selection of the 152 business units.`

From table 1 results that there are 62 women and 91 men (you said that there are 61 and 92), in total 153 respondents. Please explain the composition of the sample: 1 respondent from each business unit? In another way? We know that there is a difference between the observation unit and the survey unit. So, what is the connection between the two, in the case of your research?

- Please specify the period in which the data were collected and how did you approach the respondents (online or face to face). How did we control the quality of the data?…

The conclusions section should follow the results section. Also, this section needs to be developed, especially regarding the practical implications of including the age and gender variables in the model. Please clarify what are the theoretical and practical contributions of your research.

Greater attention to punctuation marks and technical editing (missing letters, or mistyping, e.g Fronell-Larcker Criterion. It is Fornell) is necessary. Another round of spellchecking by a native speaker is recommended.

Author Response

All comments have been attended to. See attachment. 

Reviewer 4 Report

The authors incorporated the Technology Acceptance Model (TAM) and Innovation Diffusion Theory (IDT) theories to identify the factors affecting SMEs' Social Media Marketing use. The major revision items for your manuscript as below. I will attempt to summarize my concerns next, and I hope this is useful for the authors to improve the manuscript.

1. In my opinion, there are significant gaps in the originality of this manuscript. The manuscript has been organized well, but the paper's originality should be emphasized more clearly. For instance, the introduction section is quite long and confusing. It could be rewritten to be much more succinct. Furthermore, it is unclear to the reader what knowledge gap, problem, or puzzle motivates the research and this manuscript. Where exactly is the gap that this paper will fill? What is the research question? What is the contribution of this paper? Moreover,  there are many claims without evidence from literature or are supported by out-of-date citations. Here the authors have to update the introduction section with a recent and relevant citation from the last three years. The authors can consider the following article as a point of reference to get an idea regarding the issue.

i. https://doi.org/10.1007/978-3-030-64987-6_16

i. http://scientiairanica.sharif.edu/article_21100.html

2. The background in the introduction could be improved by adding more related and recent references regarding Small and Medium Enterprises (SMEs) statistics and their use of social media.

3.            The most vulnerable part of this article is the theoretical underpinning, for instance, the constructs, "Perceived Usefulness (PEU)," Perceived Ease of Use (PEOU) adopted from individual theory (TAM), and construct Compatibility (COM) adapted from DOI theory which is organizational level. Here the author(s) must add more justification to support the integration between the organization and individual adoption theories. Also, why were the other constructs of DOI theory ignored? (e,g relative advantage, complexity ...and so on). The authors can consider the following article as a point of reference to get an idea regarding the issue.

https://doi.org/10.1145/3133264.3133298

4. The adopted methodology described in the manuscript needs more improvement. It needs a structured research design to present a diagram that shows the flow of research and process, and these need to be justified to demonstrate the robustness of the methodology. In addition, there are questions about how data bias was addressed during the data collection process from 152 SMEs. Also need to elaborate on the target population and report the minimum sample size required for this study using tools like G*power. These all need explicit explanations in the research method section.

5. Please add an editable version for Table 1 instead of a screenshot.

6. It is recommended to present the items used to collect the data and their sources in a new table. It can be added to the Appendix.

7.  The discriminant validity assessment is missing for this study. Need to report HTMT results.

8. The author(s) should discuss the study's findings before concluding and benchmarking the findings with the previous studies since the discussion section is shallow.

9. The theoretical and practical implications of this study should be added in new subsections, as they are missing from the current version of the manuscript.

10. Some suggestions for future research should be discussed.

Author Response

All comments have been taken care on in the revised paper

Round 2

Reviewer 1 Report

Dear Authors,

the revised article is adjusted in accordance with the comments.

Consequently, the article has been modified in such a way that I consider it suitable for publication.

Wish you good luck with the article!

Author Response

Dear reviewer,

Thank you very much for taking the time to read our work and for your helpful suggestions for making it better.

Reviewer 2 Report

I have finished the second review on the revised manuscript “Social media marketing for small and medium enterprises performance in Uganda: A structural equation model”(Manuscript ID: sustainability-1907992).

In my first review, I have identified six weaknesses for the authors to fix. In the revised manuscript, the authors only revised three of them.

(2) The authors did not adopt the whole theoretical frameworks but chose two variables from TAM and one variable from IDT. Therefore, the authors should give reasons why chose the three variables. In fact, Rogers’ model identifies perceived innovation attributes as the most important predictor of technology adoption and innovation attributes contain five variables—relative advantages, compatibility, complexity, trialability, and observability. As indicated in IDT, ease of use is similar to the concept of complexity and usefulness is one of the relative advantages.

à no justifications are provided.

(3) Rogers’ model divides all consumers into five types of adopters: innovators (the first 2.5%), early adopters (the next 13.5%), early majority (the next 34%), late majority (the next 34%), and laggards (the last 16%). Empirical studies have shown that the five types of adopters differ significantly in their demographic variables. Furthermore, Rogers’ model suggests that demographics are important variables in the early stages of an innovation’s diffusion and are capable of differentiating adopters from non-adopters. As the diffusion of an innovation comes into its late stages, demographics will no longer exert a significant effect on adoption. Empirical research shows that early adopters are more likely to be male, younger and better educated and to have higher incomes than non-adopters.

Therefore, based on Rogers’ model or TAM, demographic variables do not exert any moderating effects on the adoption. If the authors claimed that demographic variables do have moderating effects on the adoption or on the performances, then the authors should provide strong justifications for this claim.

The rebuttal from the authors

è To accept novel technical advancements, I do believe that demographics are influenced by social, environmental, and other cofounding elements that MAY not be the same across different settings (i.e., Europe, Asia, and Africa). Therefore, just because Rodgers conducted his research in one situation in which demographics had no bearing does not always mean that demographics will have no impact in all circumstances.

My comment on the authors’ rebuttal.

à Rogers’ model has been widely adopted and thus, this model has been examined in different countries, different cultures, and different areas. Same with the TAM. Therefore, if the authors believe that demographic variables exert a moderating impact on adoption, then the authors should provide justifications or empirical evidence.

(6) There is a lack of operationalization of major variables in this manuscript. Particularly, the authors need to indicate how the authors measured the variables of social media marketing and performances of small and medium enterprises. Without these data, it would be difficult for readers to estimate the validity of the two major variables.

The rebuttal from the authors

è These latent constructs (i.e., social media marketing (SMM) and small- and medium-sized business performances (POB)), can often only be quantified indirectly by analyzing the effects they have on the variables that may be directly observed. In this context, observed variables are typically responses to questions in a scale. In this study, the latent constructs were made up of indicators that mirrored them. I hope this gives light to the query of operationalization of the major variables (SMM and POB) in the model

My comment on the authors’ rebuttal.

è The authors should at least provide what constitutes each of the latent constructs to allow the reviewers to estimate the validity of these latent variables.

Author Response

Dear Editor,

Find attached rev2 rebuttals.

Reviewer 4 Report

Some of the given comments were mainly addressed, but the manuscript still has some points which need improvement. Therefore, please address all the given comments seriously and answer each comment (point by point).

 1. The knowledge gap and research problem, are still unclear in the introduction section.

2. The authors claimed that the developed research model is based on the Technology Acceptance Model (TAM) and Innovation Diffusion Theory (IDT) theories in many parts of the manuscript  (e.g., Abstract lines 20-22; lines 128-133; lines 572-574), however, the authors ignored or were unable to justify the integration between the organization and individual adoption theories. Also, why were the other constructs of IDT theory ignored? (e,g relative advantage, complexity ...and so on). How can the authors claim the integration between two theories by using only one construct (e.g., Compatibility from  IDT theory)?

3. The authors need to calculate the minimum sample size required for this study  (Are 152 respondents will be enough for such a study?)

4. As the data used in this study was collected from a single source (152 SMEs in Uganda), the authors have to report the common method bias (CMB) test results.

5. We need the actual data collection tools (surveys) used to obtain the results. Otherwise, we cannot effectively judge the quality of the results. Please add the items used to collect the data and their sources in a new table.

6. The author reported the Fornell and Larcker Criterion results for assessing the discriminant validity. However, Fornell and Larcker’s Criterion is still lacking in establishing the distinctiveness between constructs, which prompted them to suggest a more robust approach that could capture the discriminancy among the constructs under study. To solve this issue, the authors have to report the results of the Heterotrait-monotrait (HTMT) criterion as a new criterion for assessing discriminant validity suggested by Henseler et al. (2015). Please refer to the following article

https://link.springer.com/article/10.1007/s11747-014-0403-8

7. The added discussion section only represents the results. Here the authors have to interpret the meaning of the results and put them in context (SMEs in Uganda), then explain why they matter.

Author Response

attached are the rebuttals of rev4

Round 3

Reviewer 2 Report

In my second review, I identified the following three weaknesses for the authors to fix.

(1) The authors did not adopt the whole theoretical frameworks but chose two variables from TAM and one variable from IDT. Therefore, the authors should give reasons why chose the three variables. In fact, Rogers’ model identifies perceived innovation attributes as the most important predictor of technology adoption and innovation attributes contain five variables—relative advantages, compatibility, complexity, trialability, and observability. As indicated in IDT, ease of use is similar to the concept of complexity and usefulness is one of the relative advantages.

The rebuttal from the authors

The use of some TAM components is justified in Section 2.1, paragraph one. Additionally, Section 2.1 paragraph three makes a strong case for why only some of the IDT model parts should be used.

My comments:

I do not consider the justification strong to be convincing, but I can accept the rationales provided. However, according to these rationales, the authors adopted TAM as the theoretical framework for this manuscript and borrowed one variable—compatibility—from Rogers’ diffusion of innovation model. If TAM is the theoretical framework for this manuscript, then the conceptual model should be as follows, not the conceptual model in figure 1 of page 7. Therefore, the authors need to re-conduct the SEM to ensure the relationships among the three sets of variables. That is, social media marketingà compatibility, perceived usefulness, and perceived ease of useà performance of business.

(2) Rogers’ model divides all consumers into five types of adopters: innovators (the first 2.5%), early adopters (the next 13.5%), early majority (the next 34%), late majority (the next 34%), and laggards (the last 16%). Empirical studies have shown that the five types of adopters differ significantly in their demographic variables. Furthermore, Rogers’ model suggests that demographics are important variables in the early stages of an innovation’s diffusion and are capable of differentiating adopters from non-adopters. As the diffusion of an innovation comes into its late stages, demographics will no longer exert a significant effect on adoption. Empirical research shows that early adopters are more likely to be male, younger and better educated and to have higher incomes than non-adopters.

Therefore, based on Rogers’ model or TAM, demographic variables do not exert any moderating effects on the adoption. If the authors claimed that demographic variables do have moderating effects on the adoption or on the performances, then the authors should provide strong justifications for this claim.

The rebuttal from the authors

Although I think this was addressed earlier, I can shed further light on it. I must make the reviewer aware of the following: “research is not a one size fits all’’. There will never be remarks like "What was once thought to be impractical decades ago does not inevitably mean that it will or will never work elsewhere", “on-going research or rather further research is needed….”.

We gathered data using a scientific procedure that was clearly explained, and we feel that errors were considerably reduced. We then evaluated the data and reported the findings. We WILL NOT change the findings to match those made elsewhere or under different economic circumstances.

Therefore, based on Rogers’ model or TAM, demographic variables do not exert any moderating effects on the adoption.” It is scientifically incredible to accept this discovery as being universal. In various communities, particularly in African and Western cultures, the condition and behavior of demographics like gender and age are extremely varied.

The reviewer is also urged to read the work:

Chawla, D., & Joshi, H. (2018). The moderating effect of demographic variables on mobile banking adoption: an empirical investigation. Global Business Review19(3_suppl), S90-S113.

https://doi.org/10.1177/0972150918757883

May be to add on what the reviewer pointed out, stating that, these models have been applied in different settings etc.…… Along with reporter integrity, sample size is another important factor to take into account. Perhaps a new study might be conducted to examine how sample size influences the findings in this specific demographics scenario. I must say that some researchers manipulate findings to make claims that reviewers will find interesting.……a case in point is this one we are on.

My comments:

Maybe the authors have different points with mine. Let me set the differences aside and pay attention to the regression analyses the authors did to analyze the moderating effects of gender and age., which are summarized in Table 7, Table 8, Table 9, and Table 10.

(a) For Table 7 that summarizes the results of the moderating effect of gender on POB_AV, the correct regression analysis should be as follows, not the one in Table 7 of page 14. In this regression analysis, the authors will be able to understand how gender interacts with each of the three variables—PEOU_AV, COM_AV, and FEU_AV

to affect POB_AV.

 (b) For Tables 8-10 that summarize the results of the moderating effect of age on POB_AV, the authors should treat the variable of age as a continuous variable rather than dividing the category of age into three groups—18-23, 24-29, and 30-35. Furthermore, the correct regression analysis should be as follows, not the one in Table 8, Table 9, or Table 10. Similarly, in this regression analysis, the authors will be able to understand how age interacts with each of the three variables—PEOU_AV, COM_AV, and FEU_AV to affect POB_AV.

(c) In addition, the authors stated that H5 was supported under the age category “18-23”. However, the data in Table 8 showed that there was no significant effects of interaction because the p value was .079.

(3) There is a lack of operationalization of major variables in this manuscript. Particularly, the authors need to indicate how the authors measured the variables of social media marketing and performances of small and medium enterprises. Without these data, it would be difficult for readers to estimate the validity of the two major variables.

The rebuttal from the authors

Table 2 displays the outcomes of the measured model. Unless the reviewer requests the variable labels rather than the variable names that were utilized in the report.

My comments:

When I looked at Table 2 that showed, for example, performance of business contained POB1, POB2, POB3, POB4, and POB5, I still would not be able to estimate the congruence between conceptualization and operationalization of variables. This is important because the congruence between the two allows me to measure the validity of this manuscript, which is one type of validity measurement.

Author Response

Thank you so much for your insights that have made us improve our manuscript.

Reviewer 4 Report

The authors ignored or were unable to address most of the given comments, for example

1. The greatest problem with the developed model in this study is related to the proposed model's novelty. The authors ignored or were unable to justify the integration between the Technology Acceptance Model (TAM) and Innovation Diffusion Theory (IDT) theories. Why only one variable from IDT theory which is “Compatibility” is enough?

2.  Are 152 respondents will be enough for such a study?) need to justification!

3. The common method bias (CMB) test result is required because the data used in this study were collected from a single source (SMEs in Uganda).

4. We need the actual data collection tools (surveys) used to obtain the results. Otherwise, we cannot effectively judge the quality of the results. Please add the items used to collect the data and their sources in a new table. In this comment, we asked the authors to add the items (questions) used in the survey and their sources.

5. The author reported the Fornell and Larcker Criterion results for assessing the discriminant validity. However, Fornell and Larcker’s Criterion is still lacking in establishing the distinctiveness between constructs, which prompted them to suggest a more robust approach that could capture the discriminancy among the constructs under study. To solve this issue, the authors have to report the results of the Heterotrait-monotrait (HTMT) criterion as a new criterion for assessing discriminant validity suggested by Henseler et al. (2015). Please refer to the following article

https://link.springer.com/article/10.1007/s11747-014-0403-8

6. The subsections “6.1 Theoretical contributions”, “6.2 Practical Implications”, and “8. Limitations and Directions for future research” need to move under the “Conclusion” section. Also, pls check the numbering of sections and sub-sections, and correct the mistake.

Author Response

I have no comment.